# Antitumoral and Immunogenic Capacity of β-D-Glucose—Reduced Silver Nanoparticles in Breast Cancer

**DOI:** 10.3390/ijms24108485

**Published:** 2023-05-09

**Authors:** Pedro Félix-Piña, Moisés Armides Franco Molina, Diana Ginette Zarate Triviño, Paola Leonor García Coronado, Pablo Zapata Benavides, Cristina Rodríguez Padilla

**Affiliations:** Laboratorio de Inmunología y Virología, Facultad de Ciencias Biológicas, Universidad Autónoma de Nuevo León, San Nicolás de los Garza 66455, NL, Mexico

**Keywords:** nanotechnology, β-D-glucose, silver nanoparticles, breast, cancer, immunogenic cell death

## Abstract

Immunogenic cell death (ICD) is a type of cell death capable of stimulating immunity against cancer through danger signals that lead to an adaptive immune response. Silver nanoparticles (AgNPs) have been shown to have a cytotoxic effect on cancer cells; however, their mechanism of action is not fully understood. The present study synthesized, characterized, and evaluated the cytotoxic effect of beta-D-glucose-reduced AgNPs (AgNPs-G) against breast cancer (BC) cells in vitro; and assess the immunogenicity of cell death in vitro and in vivo. The results showed that AgNPs-G induce cell death in a dose-dependent manner on BC cell lines. In addition, AgNPs show antiproliferative effects by interfering with the cell cycle. Regarding the detection of damage-associated molecular patterns (DAMPs), it was found that treatment with AgNPs-G induces calreticulin exposure and the release of HSP70, HSP90, HMGB1, and ATP. In vivo, prophylactic vaccination did not prevent tumor establishment; however, tumor weight was significantly lower in AgNPs-G vaccinated mice, while the survival rate increased. In conclusion, we have developed a new method for the synthesis of AgNPs-G, with in vitro antitumor cytotoxic activity on BC cells, accompanied by the release of DAMPs. In vivo, immunization with AgNPs-G failed to induce a complete immune response in mice. Consequently, additional studies are needed to elucidate the mechanism of cell death that leads to the design of strategies and combinations with clinical efficacy.

## 1. Introduction

Breast cancer (BC) is one of the most frequently diagnosed neoplasms worldwide and is one of the leading causes of death from cancer [1]. BC can be classified based on genetic/hormone characteristics, as the estrogen receptor, progesterone receptor, and the human epidermal growth factor 2 receptor (HER2) expression, being the absence of all mentioned the most aggressive phenotype, known as triple negative BC [2]. Although the standard treatment against BC involves a multidisciplinary approach that includes surgery, radiotherapy, and neoadjuvant/adjuvant systemic therapies [3,4] that increases relative survival rate up to 5 years, numerous patients still suffer from recurrence considered a major obstacle [5]. Recent advances in cancer immunotherapy have enabled innovative BC treatment, to overcome the immune system evasion that certain tumors exhibit by limiting the processing and antigen presentation in cold microenvironments. For these, is necessary to develop a better innovative immunotherapy that prolongs patients’ survival. [6]; to resolve this problem, various strategies have been developed to induce immunogenic cell death (ICD), an example of this is the use of chemotherapy treatments, such as doxorubicin, that has been demonstrated to be efficient in preclinical studies [7]. Leaving a research gap yet to be proven in a clinical stage.

The ICD requires the death of cancer cells so that it can induce long-lasting antitumor immunity [8]. This scenario involves a specific T cell attack dependent on the participation of antigen-presenting cells and alarmins, such as calreticulin (CRT) in the cell surface, the exposure and release of heat shock proteins 70 and 90 (HSP70 and HSP90, respectively), and the release of ATP, box 1 of the high mobility group of non-histone chromatin protein (HMGB1) [9], and presence of the tumor-specific antigens to achieve a specific response. To establish a treatment inductor of ICD, in vivo prophylactic vaccination assays derived from treated cancer cells remain the gold standard [10].

In breast cancer, the clinical prognosis related to survival associated with a treatment that induces ICD is not yet established; however, an ICD-linked prognostic signature model was developed and verified, demonstrating an association between overall survival of the breast invasive carcinoma patients and tumor immune microenvironment, concluding in a novel ICD-based breast invasive carcinoma classification scheme [11]. These findings are relevant but should be demonstrated in a clinical setting. In BC, the identification of biomarkers of early translocation, such as CRT, the exposure and release of HSP70 and HSP90, ATP, and HMGB1 [12] linked to the ICD could be attributed to the benefit from immunotherapy, and treatments with capacity to induce ICD. 

Colloidal silver induces a strong cytotoxic activity against MCF-7 cancer cells [13], such as silver nanoparticles, on CT26 mouse colon carcinoma and MCA205 mouse fibrosarcoma cell lines [14], generating a growing interest in cancer treatment using these materials. Nonetheless, in last mentioned the antitumoral activity was demonstrated but without the ability to induce ICD in CT26 mouse colon carcinoma and MCA205 mouse fibrosarcoma models. Despite these results, it is indispensable to corroborate the effect of the diverse types of synthesis of silver nanoparticles since different processes show variability in the size, morphology, antitumor effect, and ability to induce ICD. The use of β -D-glucose as a reducing agent in the production of AgNPs (AgNPs-G) in recent studies has been demonstrated to be a stabilizer that controls the growth, morphology, electrical charge, dispersion, and dissolution (release of ions) of AgNPs, exacerbating their cytotoxicity on cancer cells [15]. The present study aims to determine the cytotoxic effect of AgNPs-G on mouse and human breast cancer cell lines and their ability to release alarmins that could induce ICD.

## 2. Results

### 2.1. AgNPs-G Characterization 

The first step was to develop a chemical synthesis for the formation of stable AgNPs-G. The generation of AgNPs-G was confirmed by UV-visible spectroscopy showed at day 0 a maximum absorbance band at a wavelength of 423 nm (Figure 1A), characteristic of silver nanoparticles and this spectrum was maintained until day 80 (Figure 1B).

We observed the synthesis process reproducibility based in five independent measurements at day 0 (Figure 1C). The size distribution of AgNPs-G ranged between 0.1 and 10 nm with an average size of 5.991 nm and a polydispersity of 0.228 (Figure 1 D). The zeta potential of AgNPs-G was −7.48 mV (Figure 1E). Morphology and size were corroborated by TEM and AFM. TEM micrographs suggest sizes between 0.1 and 35 nm with averages between 0.1 and 5 nm (Figure 2A). 

TEM micrography showed a quasi-spherical shape (Figure 2B). AFM images reveal topographical details of the nanoparticle surface corroborating this morphology (Figure 2C). The elemental composition of the AgNPs-G was determined by XPS. The XPS survey exhibits the presence of Ag, O, and C (Figure 2D). High-resolution XPS spectra at the Ag(3d) electron core level showed two symmetrical peaks at 366.8 and 372.78 eV consistent with Ag(3d_5/2_) and Ag(3d_3/2_), respectively (Figure 2E). The separation between the peaks of the Ag(3d) region was 6.0 eV.

### 2.2. Effect of AgNPs-G on the Viability and Cell Cycle of BC Cells

We evaluated the effect of different concentrations of AgNPs-G on non-tumorigenic epithelial cell line MCF 10A, breast cancer cell lines, and PBMC. As shown in Figure 3A, AgNPs-G treatment induced a dose-dependent decrease in all cell lines viability compared to the control. 

CC50 values for each cell line were 78 μM for 4T1, 67 μM for MCF-7, 46 μM for MDA-MB-231, 71 μM for SKBR-3, 36 μM for MCF-10A, and 85 μM for PMBC. Subsequently, we investigated whether the decrease in cell viability induced by AgNPs-G treatment was related to cell cycle arrest except for MCF-10A and PMBC because were only used as controls for cytotoxic specificity assessment. As can be seen in Figure 3B, the progression of the cell cycle was dependent on the cell line. Treatment with AgNPs-G increased the percentage of cells in the SubG1 phase in the 4T1 (5.8%), MCF-7 (7.6%), and SKBR-3 (18.2%) lines compared to the untreated group. The G1 phase decreased in cell lines MCF-7 (12.6%), MDA-MB-231 (1.97%), and SKBR-3 (2.8%) treated with AgNPs-G. The synthesis phase decreased in lines 4T1 (7.2%) and SKBR-3 (21.1%) after treatment with AgNPs-G.

### 2.3. Treatment with AgNPs-G Induces DAMPs Exposure and Release in BC Cells

Once the cytotoxic potential of AgNPs-G against BC lines was determined, we evaluated DAMPs exposure and release. The release of HMGB1 varied according to the cell line and the concentration of AgNPs-G. At CC50, HMGB1 release was significantly increased in all cell lines except the MCF-7 cell line. On the other hand, the release of HMGB1 in CC100 was considerably increased in all cell lines compared to the control (Figure 4A). 

The release of ATP increased in almost all the cell lines after the administration of the treatments with both concentrations, only in the MCF-7 cell line at the CC50 concentration the increase was not significant compared to the control (Figure 4B). The expression of HSP70 and HSP90 decreased in the cell lysates of the lines MCF-7, MDA-MB-231, and SKBR-3 treated with AgNPs-G CC50 (Figure 5A). 

In contrast, the expression of these proteins was increased in the supernatants of cells treated with AgNPs-G CC50 compared to untreated cells (Figure 5B). Finally, the treatments with AgNPs-G CC50 presented a significant increase in calreticulin exposure in all cell lines analyzed by flow cytometry. The 4T1 cancer cell line exposed the largest quantity of calreticulin (Figure 6). 

### 2.4. Vaccination with Cells Treated with AgNPs-G Did Not Prevent Tumor Establishment

Once the effect of the AgNPs-G on cell viability, expression, and release of DAMPs in vitro was determined, the next step was to carry out a prophylactic vaccination, to determine the capacity to induce ICD. The results showed that prophylactic vaccination with cells treated with AgNPs-G CC50 (V-AgNPs-CC50) and CC100 (V-AgNPs-CC100) did not prevent tumor establishment (Figure 7A). 

However, the tumor weight was significantly lower in the vaccinated mice compared to the control (Figure 7B). Additionally, the survival rate (Figure 7C) increases in mice vaccinated with V-AgNPs-CC50 (18 days) and V-AgNPs-CC100 (16 days).

## 3. Discussion

First, we obtained silver nanoparticles reduced with β-D- glucose with a surface plasmon resonance distinctive of silver in nanometric size in a range of 400–480 nm [16,17,18]. The nanoparticles showed an average size of 5.991 nm with quasi-spherical morphology, probably due to the use of glucose as reducing agent [19], these results correlate with Panzarini et al. [20] who reported the obtention of spherical NPs coated with β-D- glucose with sizes between 20 and 40 nm. Furthermore, we obtained a −7.48 mV zeta potential value with stability for 80 days, similar to the AgNPs prepared by microwave irradiation in a study by Singh et al. 2014 [21], that demonstrated nanoparticle stability with zeta potential of −5.11 mV, and an increase in stability when the zeta potential augments at −15.3 mV. 

We could determine that the silver solution is exhibited in Ag^0^ (metallic) state based on XPS analysis, such as those found by Figueiredo et al. 2022 [22] and Du et al. 2022 [23]. The detailed regions of the XPS analysis show a simple doublet in the Ag3d spectrum with an energy separation of 6.0 eV, where we could also observe the Ag3d_5/2_ component centered at 366.8 eV. Based on the data above and considering the values dispersion reported in the literature for those is correlated with the synthesis of the metallic AgNPs [23,24].

As reported by Panzarini et al., 2017 [20]. the cytotoxic effect of AgNPs-G on breast cancer cell lines was also demonstrated. Nonetheless, a comparative effect with previous authors with our results is not completely possible to establish regarding to cytotoxic activity because of the different cancer cell lines used and the variability in the form and size of the nanoparticles obtained. Mostly, the effect of the nanoparticles is dependent on factors that involve physicochemical characteristics, such as size, shape, coating, and the presence of contaminants derived from their synthesis [24]. The cytotoxic effect of AgNPs is maintained despite differences in reducing agents used for their synthesis, silver keeps its cytotoxic effect. Similar results were obtained with the treatment of MCF-7 and HEP-2 cells with AgNPs presenting spherical shape with size in the range of 5–40 nm but reduced with natural medicinal *Piper nigrum* extract [25]. 

Our study is consistent with others, by demonstrating the sensitivity of breast cancer cells to treatment with AgNPs [26,27], especially on triple-negative breast cancer cells. This predisposition may be due to the long-chain polyunsaturated fatty acids enrichment characteristic in this type of cancer, which is prone to AgNPs-driven peroxidation [28]. The AgNPs-G affects the breast cancer cell cycle in a dose-dependent manner, interfering with their growth. This data is important because some drugs used in cancer treatment can affect the cell cycle [29]. Opening a new perspective in the study of AgNPs-G on cell cycle and checkpoint regulators and their interactions with drugs that affect this mechanism of action.

On another side, there are reports of AgNPs reduced with glucose and their capacity to induce alarmins related to ICD, however, when the gold standard is performed, the ICD is not generated [14]. This possible mechanism of action could appear if the quantity or quality of alarmins or tumor neoantigens is affected by treatment, as in B16F10 cells treated with colloidal silver / AgNPs as a prophylactic vaccine in vivo by Torres et al., 2020 and García et al., 2023, respectively [14,30], not preventing tumor implantation in mice of the C57BL/6 strain, despite alarmins presence. However, in our study, a certain grade of immunity is developed as it delays tumor growth, indicating that some mechanisms, such as mentioned ahead, can be implicated and should be clarified.

(A) The treatment affects the quality of the main 4T1 cells antigens capable to induce a specific T cell response capable of attacking 4T1 cells.

(B) A great amount of AgNPs-G is present in the 4T1 cells lysate used for prophylactic vaccination, that could not be eliminated by centrifugation, which can affect the optimal antigen presentation of dendritic cells, inducing a partial specific response of T cells.

(C) The residual AgNPs mentioned above, can affect the lymphocyte capacity to produce and release granzymes avoiding tumor death. 

However, to elucidate the induction of real ICD capacity, it is necessary to administer AgNPs-G in an in vivo tumoral model, and after a time of the tumor being eliminated, challenge mice with 4T1 cancer cells to determine if a certain degree of immunity is developed to avoid a recidivate. Similar to Tao Huand et al., 2023 [6] who demonstrates with this methodology that ICD can be restored through the granzyme B activity. 

In conclusion, we have developed a new synthesis of AgNPs-G, with antitumoral activity on breast cancer cell lines affecting their cell cycle and inducing a partial immunogenic cell death mechanism demonstrated by reducing tumor size and prolonging survival in mice with prophylactic vaccination. 

## 4. Material and Methods

### 4.1. Reagents and Antibodies

Silver nitrate (AgNO_3_) was purchased from Sigma-Aldrich (St. Louis, MO, USA). β-D-glucose was obtained from Cayman Chemical Company (Ann Arbor, MI, USA) Phycoerythrin (PE)-conjugated calreticulin monoclonal antibodies (Cat. No. ADI-SPA-601PE-F) and IgG1 isotype control (Cat. No. ADI-SAB-600PE-D) was obtained from Enzo Life Sciences (Farmingdale, NY, USA). Mouse monoclonal antibodies to HSP70 (cat. no. sc-24), β-actin (cat. no. sc-69879), and rabbit polyclonal antibody HSP90 (cat. no. sc-7947) were obtained from Santa Cruz Biotechnology (Santa Cruz, CA, USA). High Mobility Group Box 1 Protein (HMGB1) BioAssay™ ELISA Kit (mouse) was purchased from US Biological Life Science (Salem, MA, USA).

### 4.2. Cell Lines and Culture Conditions

The non-tumorigenic epithelial cell line MCF-10A (ATCC^®^ CRL-10317™), human mammary carcinoma cell lines MCF7 (ATCC^®^ HTB-22™), MDA-MB-231 (ATCC^®^ HTB-26), SK-BR3 (ATCC^®^ HTB-30), and murine 4T1 (ATCC^®^ CRL-2539) were obtained from the ATCC (American Type Culture Collection, Manassas, VA. USA). MCF-10A cells were cultured in Mammary Epithelial Cell Growth Medium (MEGM) and the rest in Dulbecco’s Modified Eagle Medium (DMEM) (GIBCO^®^, Thermo Scientific, Waltham, MA, USA), both supplemented with 10% fetal bovine serum (FBS) (GIBCO^®^, Mexico), and 1% antibiotic-antimycotic (penicillin, streptomycin and amphotericin B) (Sigma, St. Louis, MO, USA). All cell lines were maintained under specific conditions at a temperature of 37 °C, relative humidity of ≈85%, and atmosphere of 95% air and 5% CO_2._

### 4.3. Synthesis of Silver Nanoparticles Using β-D-Glucose (AgNPs-G)

The synthesis of AgNPs-G was carried out following the methodology described by Panzarini E et al. (2017) with some modifications detailed below (18). 10 mL of 0.3 M aqueous solution of β-D-glucose was placed in a beaker at 120 °C for 5 min. Subsequently, 100 μL of 2.5 mM AgNO_3_ solution and 10 μL of 0.1 M NaOH solution were added until a yellow color change was observed, indicative of the formation of AgNPs-G. 

### 4.4. Characterization of AgNPs-G

The morphological and spectral characteristics were determined by transmission electron microscopy (TEM) (FEI Titan G2, Thermo Scientific, Waltham, MA, USA), atomic force microscopy (AFM) (NT-MDT, Moscow, Russia), X-ray photoelectron spectroscopy (XPS) (ULTRA DLD, Shimadzu Ltd., Kyoto, Japan), and ultraviolet-visible analysis (Agilent 8453 UV-Vis Spectrometer, Agilent Technologies, Santa Clara, CA, USA). Particle size and polydispersity values were measured by dynamic light scattering (DLS) and phase analysis light scattering (PALS) for zeta potential was obtained by the Zetasizer Nano ZS90 (Malvern Instruments, Malvern, UK).

### 4.5. Isolation of Blood and Peripheral Blood Mononuclear Cells (PBMC)

Blood was obtained from healthy donors by venipuncture and collected in heparinized tubes. PBMC isolation was performed by density gradient centrifugation using Ficoll-Histopaque 1077 (Sigma, St Louis, MO, USA). PBMC viability was determined by trypan blue exclusion method (Sigma, St. Louis, MO, USA).

### 4.6. Cytotoxicity Assay

Additionally, 5 × 10^3^ cells per well were seeded in 96-well microplate, one plate for each cell line and incubated overnight at 37°C in an atmosphere of 95% air and 5% CO_2._ Afterward, the cells were treated with AgNPs-G concentrations ranging from 250 to 3.9 μM for 24 h. Then, the wells were washed with phosphate-buffered saline (PBS). After that, 100 μL of Alamar blue (Sigma, St. Louis, MO, USA) at 20% *v*/*v* was added, and incubated for 4 h under the conditions previously described. Fluorescence readings were performed using a Synergy HT™ spectrophotometer (Biotek Instruments, Winooski, VT, USA) at an excitation wavelength of 535 nm and an emission wavelength of 590 nm. The percentage of cytotoxicity was defined using Equation (1).
% Cytotoxicity = 100 − [(A/B) × 100](1)
where A corresponds to the average absorbance of cells with treatment and B to the average absorbance of cells that did not receive treatment.

### 4.7. Cell Cycle Analysis

Additionally, 1 × 10^6^ cells per well were seeded in 6-well plates, treated with AgNPs-G at cytotoxic concentration 50 (CC50), and incubated for 24 h. Subsequently, the cells were collected, washed, fixed in 70% *v*/*v* ethanol, and stored at 4 °C. For cell cycle analysis, cells were washed and incubated in Triton X-100 0.5% *v*/*v*, 7-amino actinomycin D (7-AAD) solution (10 μg/mL) of for 20 min in dark conditions for evaluation by flow cytometry (BD FACSCanto II Flow Cytometer, San José, CA, USA). For each analysis, 20,000 events were collected and analyzed using FlowJo software (Becton Dickinson, Franklin Lakes, NJ, USA).

### 4.8. Calreticulin Determination

Additionally, 1 × 10^6^ cells per well were seeded in 6-well plates, treated with AgNPs-G at CC50 (for each cell line) and incubated for 2 h. Subsequently, the cells were collected, washed, and incubated for 1 h in the dark with a phycoerythrin-conjugated calreticulin monoclonal antibody (1:100) and evaluated by flow cytometry (BD Accuri^TM^ C6 Flow Cytometer, San José, CA, USA). 

### 4.9. ATP Release Assay

Additionally, 5 × 10^5^ cells were treated with AgNPs-G (CC50 and CC100 for each cell line). Extracellular ATP levels were evaluated in the supernatants using the quantitative chemiluminescence detection kit (ENLITEN^®^ ATP Assay System Bioluminescence Detection Kit for ATP Measurement, PROMEGA) following the manufacturer’s instructions. Bioluminescence was determined using a Synergy HT™ spectrophotometer (Biotek instruments, Winooski, VT, USA) at 560 nm.

### 4.10. HMGB1 Release Test

For this assay, 5 × 10^5^ cells were treated with AgNPs-G (CC50 and CC100 for each cell line) for 24 h. HMGB1 levels were evaluated in the supernatants of treated and untreated cells using the HMGB1 BioAssay ELISA kit (human) for MCF-7, MDA-MB-231, and SK-BR3 cells (Biological Life Science, Salem, MA, USA), and the HMGB1 BioAssay ELISA kit (mouse) for 4T1 cells (Biological Life Science, Salem, MA, USA) following the manufacturer’s instructions. Microplate reading was performed on a spectrophotometer (Biotek instruments, Winooski, VT, USA) at 460 nm.

### 4.11. Western Blot Analysis

Breast cancer cell lines were seeded at a density of 5 × 10^5^ cells and treated with AgNPs-G (CC50 and CC100 for each cell line) for 24 h. Cells and supernatant were collected separately. The cells were washed with PBS and homogenized using the SET 2X lysis buffer supplemented with a protease inhibitor cocktail (Thermo Scientific, Waltham, MA, USA). Protein quantification was determined using the DC Protein Assay kit (Bio-Rad, Hercules, CA, USA). For electrophoresis, 50 μg of protein was placed per lane in a 12% polyacrylamide gel. Subsequently, the proteins were transferred to a nitrocellulose membrane by electroblotting and incubated with the primary antibodies for HSP70 and HSP90 for 24 h, at 4 °C. Primary antibodies were detected using a specific secondary antibody labeled with horseradish peroxidase. Finally, revealed by the enzymatic chemiluminescence substrate system (Thermo Scientific ^TM^, Waltham, MA, USA).

### 4.12. Animals

Six- to eight-week-old female BALB/c mice were used, which were kept under animal facility conditions (temperature of 25 °C, relative humidity of ≈55% and photoperiod of 12 h light/12 h darkness). All mice were provided with food and water ad libitum. All protocols were approved by the Animal Bioethics Committee of the Faculty of Biological Sciences of the Autonomous University of Nuevo León (San Nicolas de los Garza, NL, Mexico).

### 4.13. In Vivo Antitumor Vaccination

Additionally, 5 × 10^5^ 4T1 cells were treated with AgNPs-G (CC50 and CC100) and AgNO_3_ (2.5 mM) in vitro. Subsequently, the cells were collected, washed, and centrifuged at 800× *g* for 15 min. Cells were resuspended in 200 μL PBS and inoculated subcutaneously into the left flank of the mice. Seven days later, viable 4T1 cells were inoculated subcutaneously into the right flank. Tumor volume was measured daily until sacrifice and defined by Equation (2) [31].
Tumor volume: 4/3 × π × L/2 × W/2 × h/2(2)
where L corresponds to the longest side, W to the shortest side, and h to the height of the tumor.

### 4.14. Statistical Analysis

All experiments were carried out in triplicate with an analysis of variance (ANOVA) type experimental study followed by Tukey’s post hoc test using GraphPad Prism software (San Diego CA, USA). The *p* values were considered significant as follows: *p* < 0.033 (*), *p* < 0.002 (**), *p* < 0.0002(***), and *p* < 0.0001 (****). Letters (a, b, c, d, e, and f) show a significant difference (*p* < 0.05) between treatments.

## Figures and Tables

**Figure 1 ijms-24-08485-f001:**
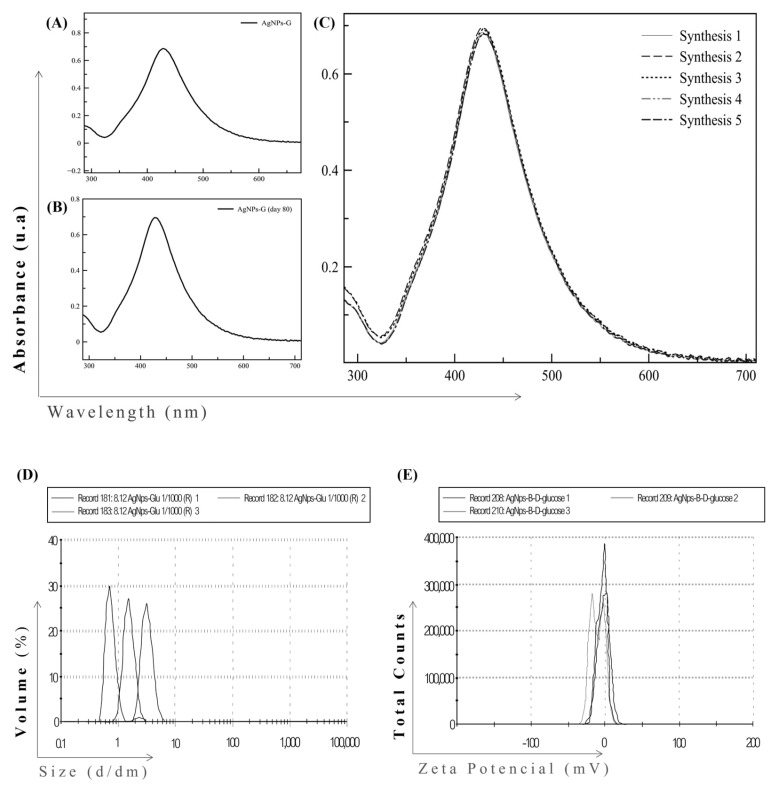
UV-Vis and DLS spectra of AgNPs-G. (**A**) UV-visible absorption spectrum of AgNPs-G at day 0, (**B**) day 80, and (**C**) synthesized five times independently. All spectra are reported as absorbance in arbitrary units (au, Y-axis) versus wavelength (nm, X-axis). (**D**) Size distribution of AgNPs-G and (**E**) zeta potential by DLS and PALS, respectively.

**Figure 2 ijms-24-08485-f002:**
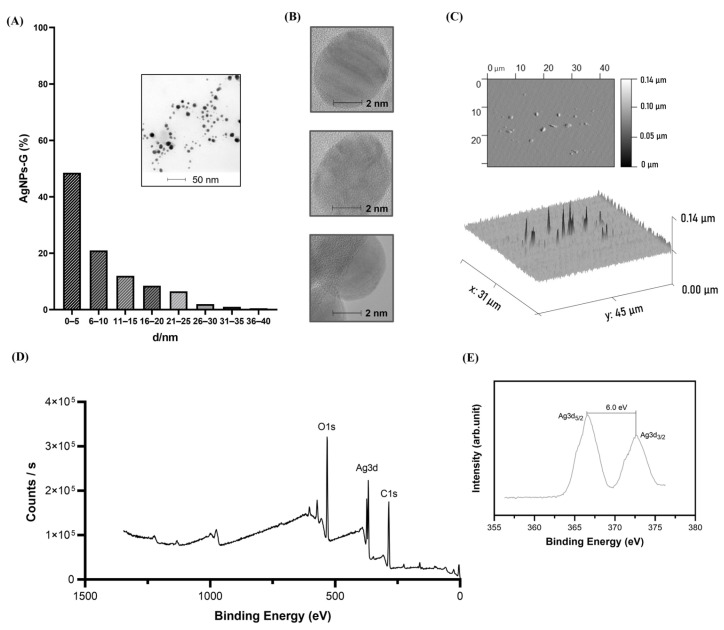
Morphological and spectral characteristics of AgNPs-G. (**A**) AgNPs-G size histogram, inset shows a typical TEM image; (**B**) TEM images of AgNPs-G; (**C**) AFM images of AgNPs-G; (**D**) XPS scan spectrum of AgNPs-G; and (**E**) high resolution XPS scan spectrum on Ag3d.

**Figure 3 ijms-24-08485-f003:**
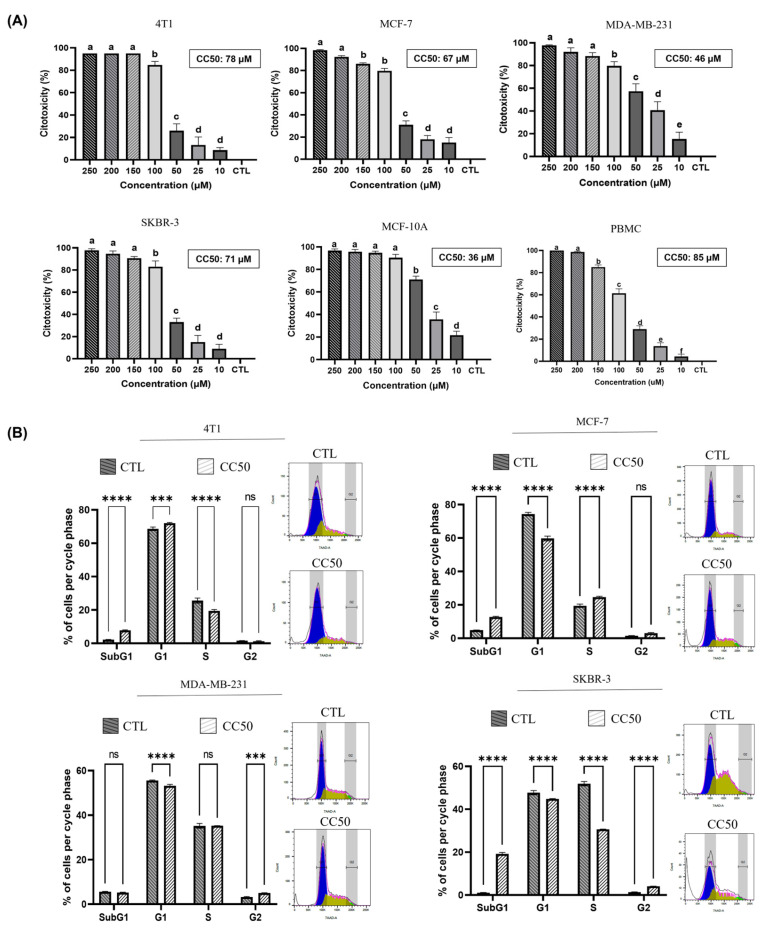
Treatment with AgNPs-G decreases cell viability and interferes with the cell cycle. (**A**) Cell viability was determined using the Alamar blue colorimetric assay on breast cancer cell lines, MCF-10A, and PBMCs. Quantitative data (mean and standard deviation) were reported. Statistical analyzes were performed by ANOVA using Tukey’s test. Letters (a, b, c, d, e, and f) show a significant difference (*p* < 0.05) between treatments. (**B**) Flow cytometric analysis for cell cycle of breast cancer cell lines treated at cytotoxic concentration 50 (CC50) and untreated (CTL). Quantitative data (mean ± standard deviation) were reported. Statistical analyzes were performed by ANOVA using Tukey’s test *p* < 0.033 (***), *p* ˂ 0.0001 (****) and ns (not significant *p* < 0.12).

**Figure 4 ijms-24-08485-f004:**
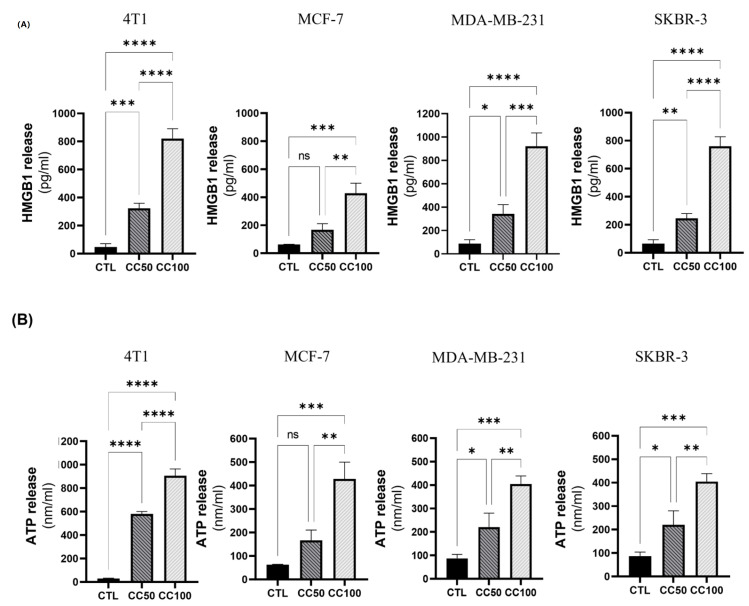
Treatment with AgNPs-G induces the release of HMGB1 and ATP. (**A**) Cells were treated at the corresponding CC50 and CC100 concentrations for each cell line. Subsequently, 100 μL of the supernatants were taken for HMGB1 measurements using an ELISA kit, according to the manufacturer’s instructions. (**B**) Cells were treated at the corresponding CC50 and CC100 concentrations for each cell line. Subsequently, 100 μL of the supernatants were taken to perform ATP measurements using a bioluminescence kit, according to the manufacturer’s instructions. Quantitative data (mean ± standard deviation) were reported. Statistical analyzes were performed by ANOVA using Tukey’s test *p* < 0.033(*), *p* < 0.002 (**), *p* < 0.0002(***), *p* ˂ 0.0001 (****) and ns (not significant *p* < 0.12).

**Figure 5 ijms-24-08485-f005:**
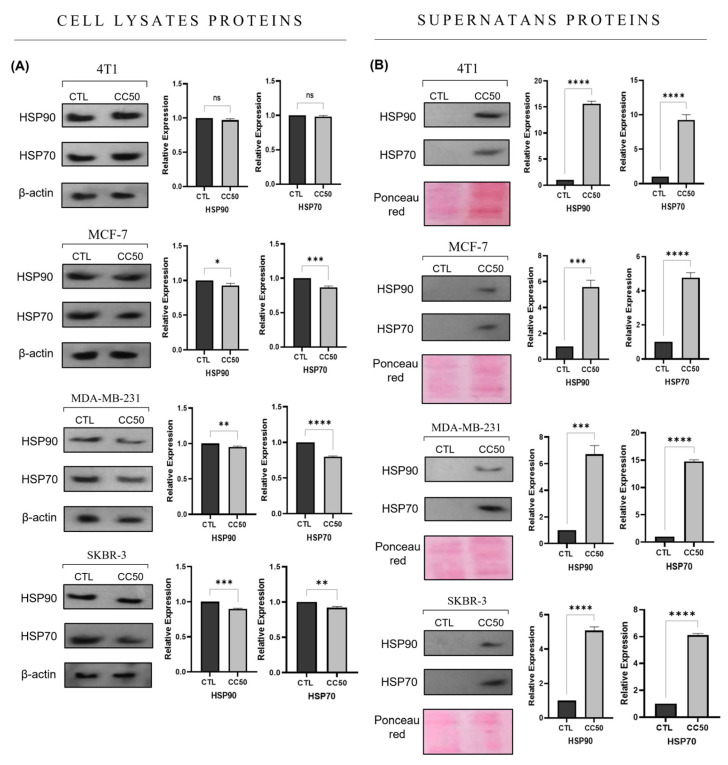
Expression and release of HSP70 and HSP90 proteins in response to treatment with AgNPs-G. The Western blot was performed with (**A**) cell lysates and (**B**) supernatants from the corresponding cell line. β-actin was used as an endogenous control. Semiquantitative analysis of the immunoblot signal was performed using Image J image analysis software to provide densitometry data for each blot. Statistical analyzes were performed by ANOVA using Tukey’s test *p* < 0.033 (*), *p* < 0.002(**), *p* < 0.0002 (***), *p* ˂ 0.0001 (****), and *ns* (not significant *p* < 0.12).

**Figure 6 ijms-24-08485-f006:**
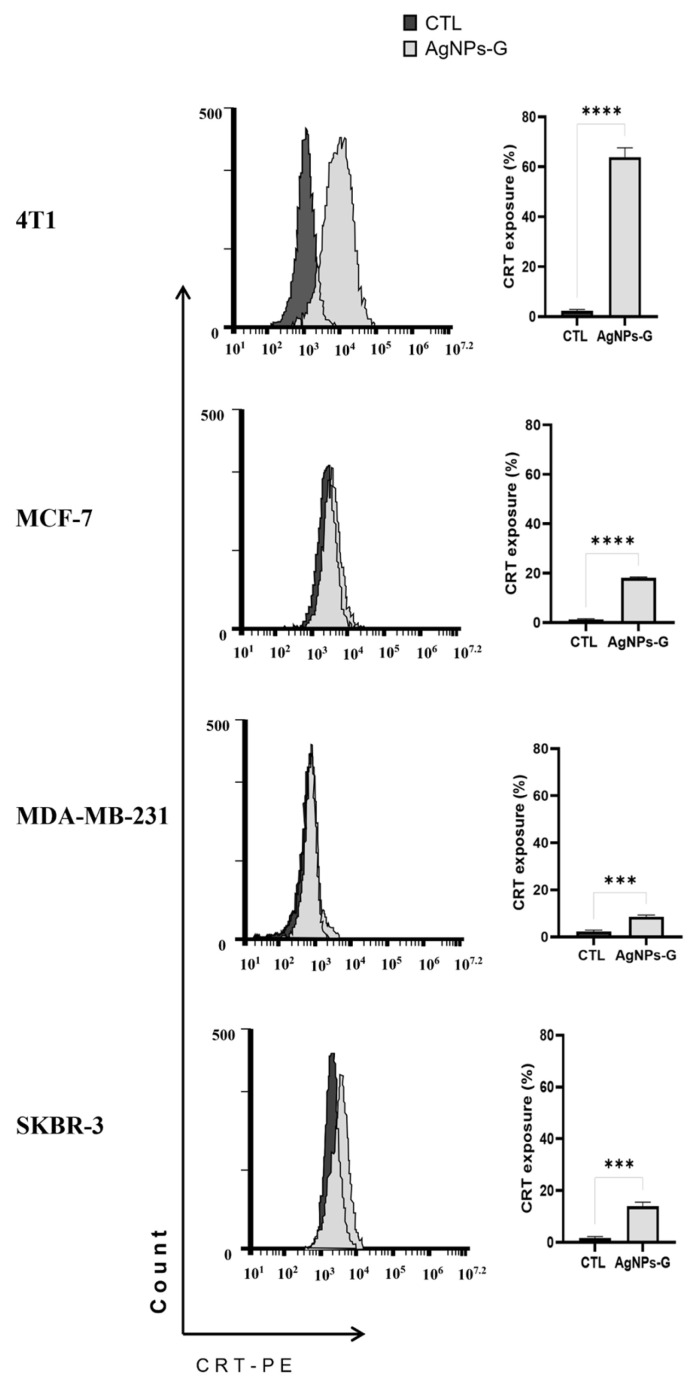
AgNPs-G treatment induced calreticulin exposure in breast cancer cell lines. The treated cells were stained using a phycoerythrin-conjugated anti-mouse monoclonal antibody to calreticulin for subsequent analysis by flow cytometry. Statistical analyzes were performed by ANOVA using Tukey’s test *p* < 0.0002 (***), and *p* ˂ 0.0001 (****).

**Figure 7 ijms-24-08485-f007:**
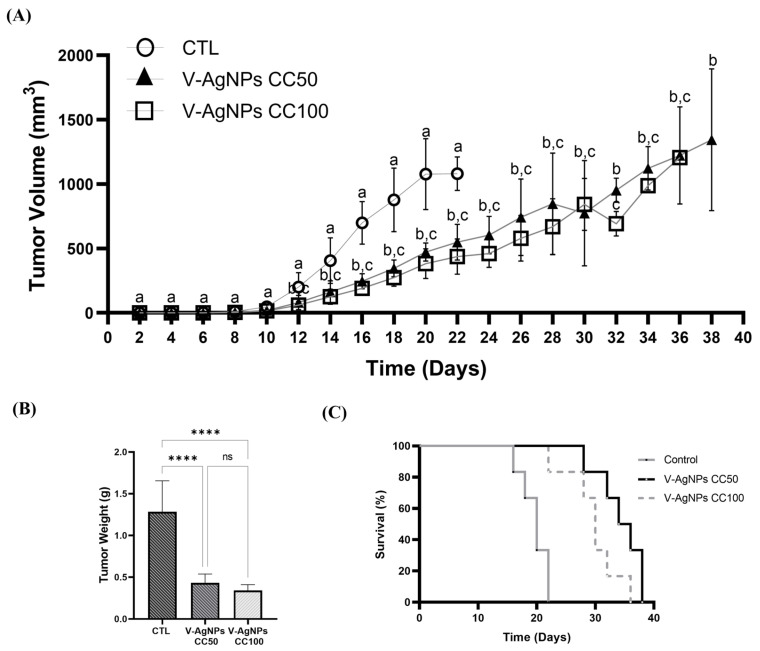
Vaccination with cells treated with AgNPs-G CC50 and CC100 did not prevent tumor establishment. (**A**) The graph indicates tumor growth in unvaccinated (CTL) and vaccinated mice with 2 × 10^6^ 4T1 cells treated with AgNPs-G CC50 and CC100. Letters (a, b, and c) show a significant difference (*p* < 0.05) between treatments. (**B**) Tumor weight. (**C**) Survival of mice represented by the Kaplan–Meier plot. Statistical analyzes were performed by ANOVA using Tukey’s test *p* ˂ 0.0001 (****), and *ns* (not significant *p* < 0.12).

## Data Availability

Not applicable.

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
