# Peer review of "Antitumoral and Immunogenic Capacity of β-D-Glucose—Reduced Silver Nanoparticles in Breast Cancer"

_ijms, 2023, doi:10.3390/ijms24108485_

Round 1

Reviewer 1 Report

General comments:

The present manuscript focuses on the role of silver nanoparticles on synthesis, cytotoxicity, and immunity against cancer through danger signals that lead to an adaptive immune response. The present study synthesized, characterized, and evaluated the cytotoxic effect of beta-D-glucose-reduced AgNPs (AgNPs-G) against breast cancer (BC) cells in vitro; and assess the immunogenicity of cell death in vitro and in vivo. The results showed that AgNPs-G induces cell death in a dose-dependent manner on BC cell lines. AgNPs show antiproliferative effects by interfering with the cell cycle. The authors developed a new method for the synthesis of AgNPs-G, with in vitro antitumor cytotoxic activity on BC cells, accompanied by the release of DAMPs. In vivo, immunization with AgNPs-G failed to induce a complete immune response in mice. The manuscript holds scientific potential but before final publication, some suggestions need to be addressed to improve the overall quality of the manuscript.

Moderate English editing is required.

Suggestions for authors:

Keywords: Follow the same pattern for all keywords.  Either the first letter should be capital or not.

Title: Modify the title a little bit. It seems more like a review.

Introduction: It is quite short. Also, there should be a connection between different subsections to highlight the importance of the current study.

Figure 1: It should be of high resolution.

Figure 5: The quality of western blots is blurred. Insert figures of high resolution.

References should be as per the journal’s format. It should be followed throughout the manuscript.

Conclusions: It is missing. It should be precise and must highlight the overall findings of the present manuscript.

Moderate English editing is required.

Author Response

TO WHOM IT MAY CONCERN:

After carrying out the review of our article proposal, based on the suggestions made by the reviewers, we have proceeded to send it for a new evaluation.

     We want to express our most sincere gratitude to the reviewers for their work, their annotations have allowed us not only to significantly improve the manuscript.

REVIEWER 1

  1. TITLE

Modify the title a little bit. It seems more like a review.

We appreciate the suggestions and have improved the title from:

β-D-GLUCOSE REDUCED SILVER NANOPARTICLES IN BREAST CANCER: SYNTHESIS, CYTOTOXICITY, AND IMMUNOGENIC PROPERTIES.

To the newly revised title of:

ANTITUMORAL AND IMMUNOGENIC CAPACITY OF β-D-GLUCOSE-REDUCED SILVER NANOPARTICLES IN BREAST CANCER

  1. KEYWORDS

Follow the same pattern for all keywords. Either the first letter should be capital or not.

The changes have been applied to the keywords and it is now in a homogeneous format.

Nanotechnology, β-D-glucose, silver nanoparticles, Breast, cancer, Immunogenic cell death.

New version:

Nanotechnology, β-D-glucose, Silver nanoparticles, Breast, Cancer, Immunogenic cell death.

  1. INTRODUCTION

It is quite short. Also, there should be a connection between different subsections to highlight the importance of the current study.

We have made the corresponding changes to the introduction according to the comments, new sections were added and are highlighted in yellow. Please find the new version below.

     Breast cancer (BC) is one of the most frequently diagnosed neoplasms worldwide and is one of the leading causes of death from cancer [1]. BC can be classified based on genetic/hormone characteristics, as the estrogen receptor, progesterone receptor and the human epidermal growth factor 2 receptor (HER2) expression, being the absence of all mentioned the most aggressive phenotype, known as triple negative BC [2]. Although the standard treatment against BC involves a multidisciplinary approach that includes surgery, radiotherapy, and neoadjuvant/adjuvant systemic therapies [3,4] that increases relative survival rate up to 5 years, numerous patients still suffer from recurrence considered a major obstacle [5]. Recent advances in cancer immunotherapy have enabled innovative BC treatment, to overcome the immune system evasion that certain tumors exhibit by limiting the processing and antigen presentation in cold microenvironments. For these, is necessary to develop a better innovative immunotherapy that prolongs patients’ survival. [6]; to resolve this problem various strategies have been developed to induce immunogenic cell death (ICD), an example of this is the use of chemotherapy treatments, such as doxorubicin, that has been demonstrated to be efficient in preclinical studies [7]. Leaving a research gap yet to be proven in a clinical stage.

The ICD requires the death of cancer cells so that it can induce long-lasting antitumor immunity [8]. This scenario involves a specific T cell attack dependent on the participation of antigen-presenting cells and alarmins such as calreticulin (CRT) in the cell surface, the exposure and release of heat shock proteins 70 and 90 (HSP70 and HSP90, respectively), and the release of ATP, box 1 of the high mobility group of non-histone chromatin protein (HMGB1) [9], and presence of the tumor-specific antigens to achieve a specific response. To establish a treatment inductor of ICD, in vivo prophylactic vaccination assays derived from treated cancer cells remain the gold standard [10].

In breast cancer the clinical prognosis related to survival associated with a treatment that induces ICD is not yet established, however, an ICD-linked prognostic signature model was developed and verified, demonstrating an association between overall survival of the breast invasive carcinoma patients and tumor immune microenvironment, concluding in a novel ICD-based breast invasive carcinoma classification scheme [11]. These findings are relevant but should be demonstrated in a clinical setting. In BC, the identification of biomarkers of early translocation such as CRT, the exposure and release of HSP70 and HSP90, ATP, and HMGB1 [12] linked to the ICD could be attributed to the benefit from immunotherapy, and treatments with capacity to induce ICD.

Colloidal silver induces a strong cytotoxic activity against MCF-7 cancer cells [13] like silver nanoparticles on CT26 mouse colon carcinoma and MCA205 mouse fibrosarcoma cell lines [14] generating a growing interest in cancer treatment using these materials. Nonetheless, in last mentioned the antitumoral activity was demonstrated but without the ability to induce ICD in CT26 mouse colon carcinoma and MCA205 mouse fibrosarcoma models. Despite these results is indispensable to corroborate the effect of the diverse types of synthesis of silver nanoparticles since different processes show variability in the size, morphology, antitumor effect, and ability to induce ICD. The use of β -D-glucose as a reducing agent in the production of AgNPs (AgNPs-G) in recent studies has been demonstrated to be a stabilizer that controls the growth, morphology, electrical charge, dispersion, and dissolution (release of ions) of AgNPs, exacerbating their cytotoxicity on cancer cells [15]. The present study aims to determine the cytotoxic effect of AgNPs-G on mouse and human breast cancer cell lines and their ability to release alarmins that could induce ICD.

  1. FIGURES

Figure 1: It should be of high resolution

We fully agree with the reviewer and have made the suggested change. The new figure is presented below for your review.

Figure 5: The quality of western blots is blurred. Insert figures of high resolution

Thanks for this suggestion, we have made the change as it improves the visualization of the figure. The new figure is presented below for your review.

  1. REFERENCES

References should be as per the journal’s format. It should be followed throughout the manuscript

The references have been changed accordingly to the journal format. Also, two new references were added. We thank you for your comments in improving the manuscript.

  1. CONCLUSIONS

It is missing. It should be precise and must highlight the overall findings of the present manuscript.

The conclusions were modified based in the comments please find the changes made from the previous version:

In conclusion, we have developed a synthesis of AgNPs-G, with antitumoral cytotoxic activity affecting the cell cycle and with a capacity to induce a partial ICD. For this, nanoparticles with different sizes should be obtained and determine their participation in ICD

To the new version:

In conclusion, we have developed a new synthesis of AgNPs-G, with antitumoral activity on breast cancer cell lines affecting their cell cycle and, inducing a partial immunogenic cell death mechanism demonstrated by reducing tumor size and prolonging survival in mice with prophylactic vaccination.

We appreciate the comments regarding the English language used. The manuscript has been reviewed by a native speaking person and modifies.

Reviewer 2 Report

The ms explores the effects of beta-D-glucose-reduced AgNPs (AgNPs-G) against breast cancers.

I suggest to compare the cytotoxicity data not only against blood and peripheral blood mononuclear cells (PBMC), but also on breast normal cells.

Moreover, statistical analysis shoudl better defined, also in figure legends.

few english errors to correct

Author Response

REVIEWER 2

1.CITOTOXICITY ASSAY

I suggest to compare the cytotoxicity data not only against blood and peripheral blood mononuclear cells (PBMC), but also on breast normal cells.

We appreciate this suggestion; we have already performed the assay on MCF-10A cell line. We did the assay on PBMCs to identify cytotoxicity. However, after the addition of the MCF-10A data the interpretation of the results indeed improved. In materials and methods, the culture medium needed for the growth of said cell line was included.

  1. STATICAL ANALYSIS

Statistical analysis should better defined, also in figure legends.

We thank you for your comments. We have described better the statistical analysis from the previous version:

All experiments were carried out in triplicate with an analysis of variance (ANOVA) type experimental study followed by Tukey's post-hoc test using GraphPad Prism software (San Diego CA, USA). The p values were considered significant as follows: p < 0.033 (∗), p < 0.002 (∗∗), p < 0.0002 (∗∗∗) and p < 0.0001 (∗∗∗∗).

To the new modified:

All experiments were carried out in triplicate with an analysis of variance (ANOVA) type experimental study followed by Tukey's post-hoc test using GraphPad Prism software (San Diego CA, USA). The p values were considered significant as follows: p < 0.033 (∗), p < 0.002(**), p < 0.0002(***) and p < 0.0001 (∗∗∗∗). Letters (a, b, c, d, e, and f) show a significant difference (p < 0.05) between treatments.

The figure legends were also changed and improved.

We appreciate the comments regarding the English language used. The manuscript has been reviewed by a native speaking person and modifies.
